**Subject Category:**
Biology (whole organism)

biochemistry/analytical chemistry

metabolomics, baicalin, allergic rhinitis

**Author for correspondence:**
Yubin Xu
e-mail: xuyubin1988@126.com

†These authors contributed equally to this work.

# Metabolomics analysis of baicalin on ovalbumin-sensitized allergic rhinitis rats

Saizhen Chen[1,†], Guirong Chen[2,†], Sheng Shu[1], Yubin Xu[1] and Xiande Ma[2]

[1]Taizhou Central Hospital (Taizhou University Hospital), Taizhou 318000, People's Republic of China
[2]Liaoning University of Traditional Chinese Medicine, Shenyang 110847, People's Republic of China

(iD) SC, 0000-0001-5830-4005

Allergic rhinitis (AR) is a global health problem that appears in all age groups and affects approximately 15–30% of people. Baicalin has been used for the treatment of various allergic diseases, including AR. However, the metabolic mechanisms of AR and baicalin against AR have not been systematically studied. Here, ovalbumin-sensitized AR rats were used as a model, and animal behaviour, histological analysis, enzyme-linked immunosorbent assay (ELISA) and metabolomics were used to elucidate the mechanism of baicalin for AR. The results indicated that baicalin has a protective effect on AR rats by inhibiting the release of immunoglobulin E (IgE), histamine, interleukin-1 beta (IL-1β), interleukin-4 (IL-4), interleukin-6 (IL-6) and tumour necrosis factor alpha (TNF-α). In addition, ovalbumin-induced AR included modulation of arachidonic acid, leukotriene A4 (LTA4), leukotriene B4 (LTB4), α-ketoglutaric acid, phosphatidylcholine PC (20 : 4/0 : 0), PC (16 : 0/0 : 0), citric acid, fumarate, malate, 3-methylhistidine, histamine and other amino acids that are involved in arachidonic acid, histidine metabolism, the TCA cycle and amino acid metabolism. Thus, AR could be alleviated or reversed by baicalin.

## 1. Introduction

Allergic rhinitis (AR) is a type of chronic inflammation in the respiratory tract [1]. It is a very common disease of nasal mucosa, triggered by the interaction of specific immunoglobulin E (IgE) and ubiquitous environmental proteins in sensitized patients [2]. The main symptoms of AR are sneezing, rhinorrhea, itching and nasal congestion; these symptoms are usually accompanied by other related symptoms such as nasal mucosa swelling [3,4].

Currently, AR appears in all age groups, affects a large part of the global population and is a major public health burden [5]. Epidemiologic data have shown that AR affects nearly 15–30% of people [6,7].

Current studies have predominantly concentrated on cytokines, chemokines such as interleukin-1 beta (IL-1β), tumour necrosis factor alpha (TNF-α) and interleukin-4 (IL-4), which are involved in the pathogenesis and process of allergic reactions during AR. However, the effective metabolites related to AR remain unclear [8,9]. It is imperative to explore potential metabolites to systematically describe the mechanism of AR.

Recently, the use of natural products for treating AR has been increasing worldwide; many prior trials have evaluated the effectiveness of herbal medicines for AR [10]. Baicalin, a flavonoid compound isolated from *Scutellaria baicalensis* Georgi, has been shown to have multiple pharmacological activities, including anti-allergic, anti-inflammatory and antioxidant effects [2]. Based on its anti-allergic and anti-inflammatory activity, baicalin is widely used to treat various allergic and inflammatory diseases, including asthma, atopic dermatitis and AR [11]. However, there is little information about the metabolic mechanism of baicalin on ovalbumin (OVA)-induced AR in Sprague-Dawley (SD) rats.

In this study, an untargeted metabolomics technique was used to evaluate the metabolic mechanism of AR SD rats, and the systematic mechanism of baicalin against OVA-induced AR was explored. This study might provide new insights to understand AR as well as the potential application of baicalin as a drug for AR treatment.

# 2. Material and methods

## 2.1. Reagents

Baicalin was purchased from the National Institute for the Control of Food and Drug (Beijing, China). HPLC-grade methanol, acetonitrile and formic acid were purchased from Merck (Germany). The enzyme-linked immunosorbent assay (ELISA) kits for IgE, histamine, IL-4, IL-1β, interleukin-6 (IL-6) and TNF-α were purchased from Uscn Life Science Inc. (Wuhan, China). OVA, leucine-enkephalin and all the authentic standards were purchased from Sigma-Aldrich (St. Louis, MO, USA).

## 2.2. Animal experiments and sample collection

Male SD rats (200 ± 20 g) were obtained from Liaoning Changsheng Biotechnology Co. Ltd. (Liaoning, China). The animals were maintained under specific pathogen free (SPF) laboratory conditions and received human care according to the guidelines of the Local Institutes of Health guide for the care and use of laboratory animals.

All SD rats were randomly divided into three groups, with 10 rats in each group: a control group, model group and baicalin group. First, the model group and baicalin group were sensitized by OVA using a standard protocol as described in a previous study [12]. Briefly, the rats were sensitized by OVA via injection on days 1, 5 and 10; from days 15 to 21, the sensitized rats were intranasally challenged by daily droppings with OVA. After the AR models were successfully established, the rats in the baicalin group were orally administered baicalin for 10 days from days 22 to 31 at a dose of 200 mg kg$^{-1}$, as previously reported [2]. The rats in the model group received saline on the same days. The control group rats received saline alone on the same schedule.

The numbers of sneezes and nose scratching were counted for 15 min after 2 h of the last administration. Then, after isoflurane anaesthesia, the rats were sacrificed by drawing all the blood from the abdominal aorta to obtain blood samples, and the nasal mucosa was separated and fixed in 4% paraformaldehyde at room temperature. The blood samples were centrifuged at 3000$g$ for 10 min at 4°C to obtain serum, which was then stored at −80°C until analysis.

## 2.3. Histological analysis

The nasal tissues were dehydrated, embedded in paraffin, sectioned at 4 μm thickness, stained with haematoxylin and eosin and then analysed with light microscopy.

## 2.4. Measurement of IgE, histamine, IL-1β, IL-4, IL-6 and TNF-α levels in the serum

IgE, histamine, IL-1β, IL-4, IL-6 and TNF-α levels in the serum were determined using ELISA kits according to the manufacturer's instructions.

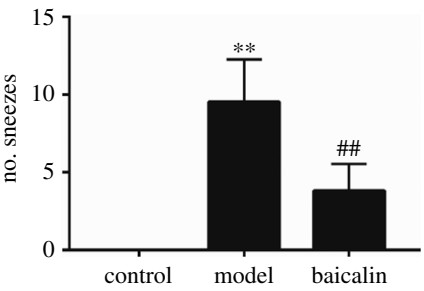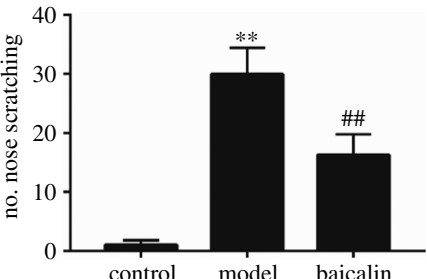

**Figure 1.** The nasal sneezes and scratching numbers of rats in each group. Compared to control group, $*p < 0.05$, $**p < 0.01$; Compared to model group, $\#p < 0.05$, $\#\#p < 0.01$.

## 2.5. Metabolomics

### 2.5.1. Sample preparation and UPLC/Q-TOF – MS/MS analysis

The method was performed as previously described with minor modifications [13]. Serum (200 µl) was added to 600 µl acetonitrile, vortex-mixed for 30 s and centrifuged at 12 000$g$ for 5 min to precipitate the proteins. Next, 600 µl of supernatant was collected and dried with vacuum drying and centrifuged at room temperature. Then, the dried residue was reconstituted with 100 µl of an acetonitrile-water solution (1 : 1, v/v). After centrifugation at 12 000$g$ for 5 min, an aliquot of 3 µl was injected for UPLC/Q-TOF–MS/MS analysis.

A Waters Acquity UPLC system was used to perform the analysis. The chromatographic column was an HSS T3 column (2.1 × 100 mm, 1.7 µm, Waters, USA), and the column temperature was maintained at 40°C. The mobile phase was methanol (A) and 0. 1% formic acid (B), and the gradient elution conditions are described as follows: 0 → 7 min: 0% → 60%A; 7 → 10 min: 60% → 100%A; 10 → 14 min: 100%A; 14 → 16 min: 100% → 0%A; Curve: 6.

Mass spectrometry was performed on a Waters Xevo G2 Q-TOF quadrupole accelerated time of flight mass spectrometer in positive and negative ion mode with an electrospray ionization (ESI) source. The desolvation gas was 800 l h$^{-1}$, the desolvation temperature was 400°C, the cone gas was 50 l h$^{-1}$ and the temperature of the ion source was 110°C. The capillary voltage, cone voltage, scanning time and collision energy were set as 3.0 kV, 45 V, 1 s and 6 eV, respectively. The spectra were collected every 0.2 s alternately with the interval at 0.02 s, and the mass range was from 50 to 1200 Da. The LockSpray$^{TM}$ was set to $m/z$ 556.2771([M + H]) and 554.2615[M − H] for leucine-enkephalin.

### 2.5.2. Data processing

The data processing was performed as previously described. Briefly, the raw MS spectra were analysed using MarkerLynx v. 4.1 (Waters Corp., UK). Then, the processed data list was exported and processed by orthogonal partial least-squares discriminant analysis (OPLS-DA) using the software package Simca v. 14.1 (Umetrics, Sweden).

## 2.6. Statistical analysis

All values were expressed as the mean ± s.d. The significance of differences between the means of control, model and baicalin groups was compared through one-way ANOVA using the Statistical Package for Social Sciences program (SPSS 20.0, USA). The significance threshold was set at $p < 0.05$ for this test.

# 3. Results and discussion

## 3.1. Animal behaviour study

The nasal symptoms of rats were recorded for 15 min. The numbers of nasal sneezes and scratching significantly increased in the model group and significantly decreased in the baicalin group. The scores for the groups are shown in figure 1. The results indicated that the OVA-sensitized rat model

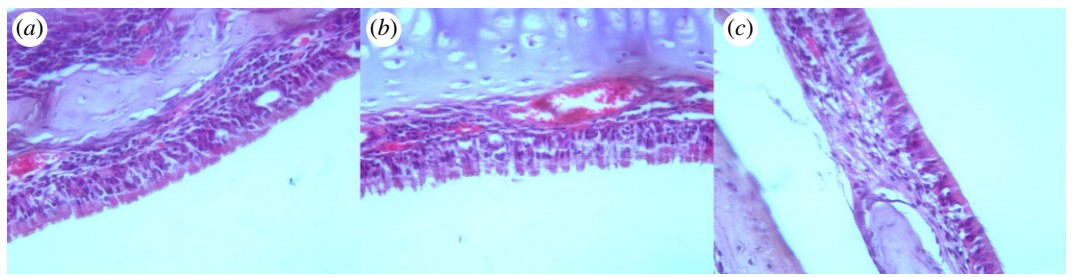

**Figure 2.** The pathological changes of nasal mucosa of rats in each group by histological analysis. (a) Control group; (b) model group; (c) baicalin group.

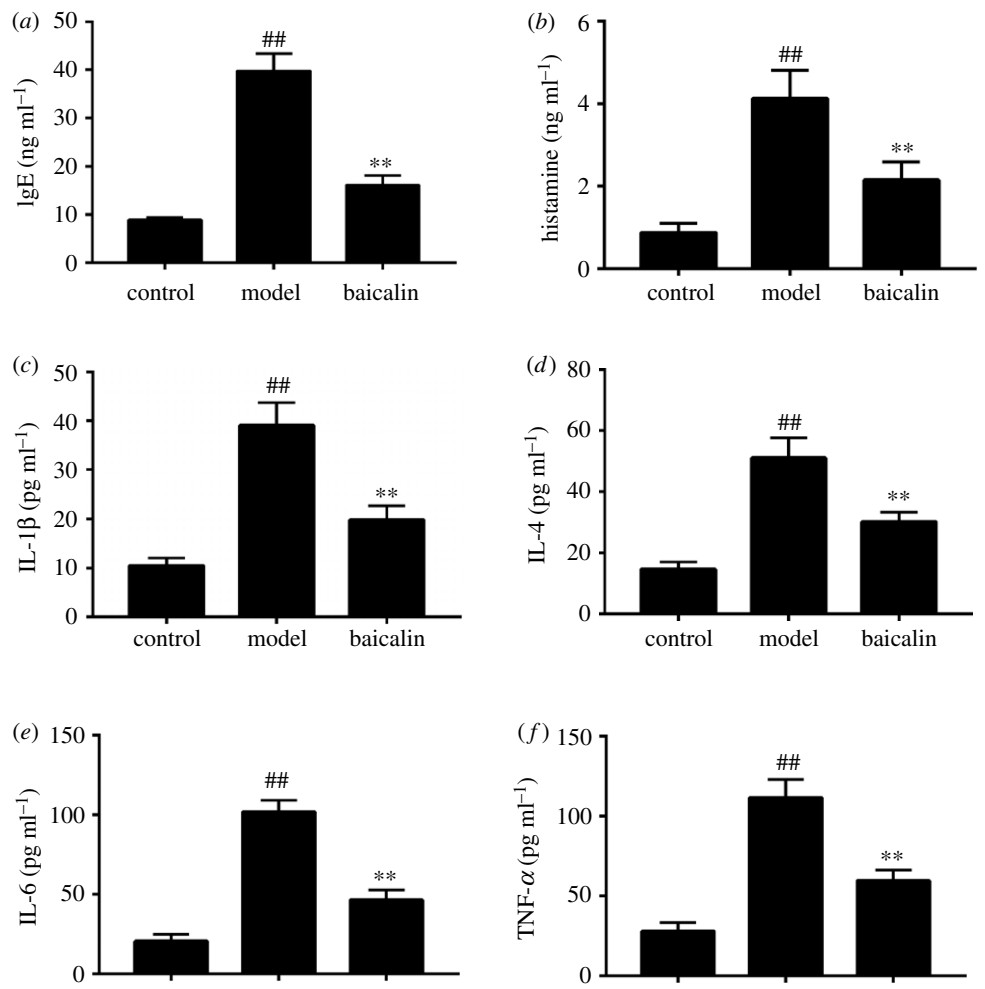

**Figure 3.** Effects of baicalin on IgE, histamine, IL-1β, IL-4, IL-6 and TNF-α level in serum. (a) IgE; (b) histamine; (c) IL-1β; (d) IL-4; (e) IL-6; (f) TNF-α. The data were presented as mean $\pm$ s.d., n = 10 per group. Compared to control group, *$p < 0.05$, **$p < 0.01$; Compared to model group, #$p < 0.05$, ##$p < 0.01$.

was successfully established, and the nasal symptoms could be inhibited by baicalin at a dose of 200 mg kg$^{-1}$.

## 3.2. Histological analysis

Histological analysis was used to evaluate the pathological changes in the nasal mucosa of rats in each group. The nasal mucosa was arranged neatly and had no abnormality in the control group, while many eosinophilic granulocytes infiltrated the nasal mucosa, and blood stasis was clearly observed in the model group. The pathological changes were significantly alleviated by baicalin (figure 2).

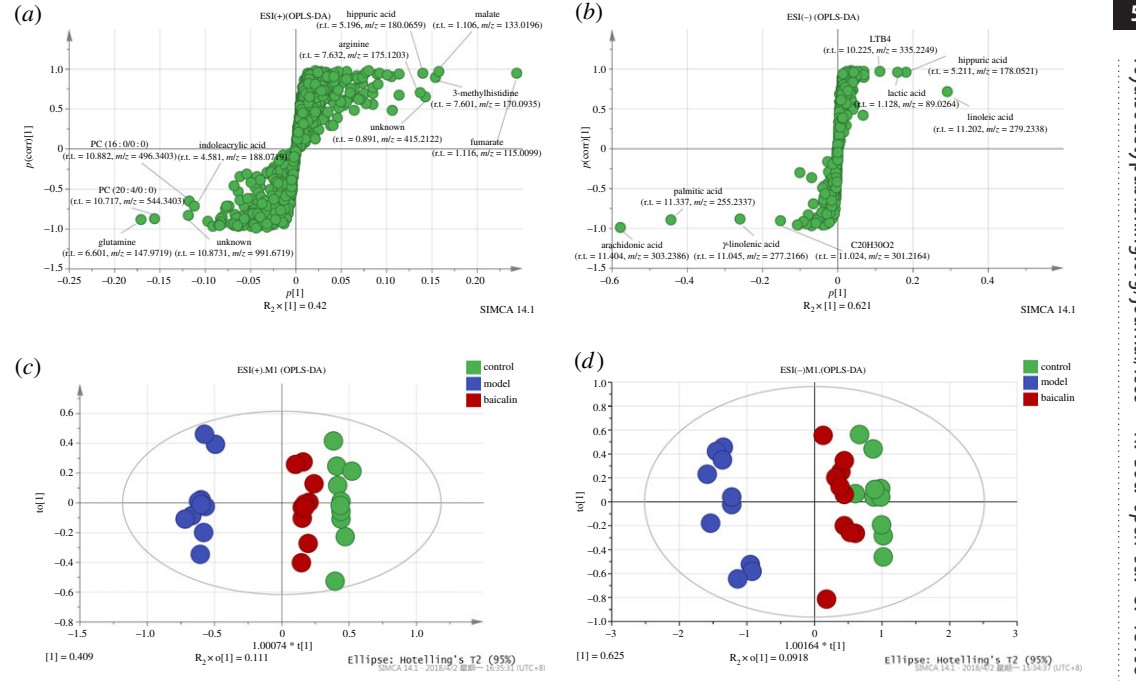

**Figure 4.** The S-plot analysis and OPLS-DA analysis. (*a,b*) The S-plot analysis between control and model groups for the metabolomics data of different ion mode. (*c,d*) The OPLS-DA analysis among control, model and baicalin groups for the metabolomics data of different ion mode.

## 3.3. Effects of baicalin on IgE, histamine, IL-1β, IL-4, IL-6 and TNF-α levels in serum

To determine the effect of baicalin on AR models, the levels of IgE, histamine, IL-1β, IL-4, IL-6 and TNF-α were measured by ELISA. The levels of IgE, histamine, IL-1β, IL-4, IL-6 and TNF-α in the model group were markedly increased compared with the control group, and $200 \, \text{mg kg}^{-1}$ baicalin significantly decreased the levels of IgE, histamine, IL-1β, IL-4, IL-6 and TNF-α in the OVA-sensitized rats (figure 3).

AR is caused by an IgE-mediated inflammatory reaction, including early- and late-phase allergenic mediators [14]. Histamine, as the most iconic indicator of vascular vitality in allergic reactions, is one of the early-phase allergenic mediators; however, histamine as a detection index still has some limitations, mainly reflected in the short half-life [15,16]. Late-phase allergenic mediators, including IL-1β, IL-4, IL-6 and the proinflammatory cytokine TNF-α, can induce IgE synthesis and play an essential role in AR development [17,18].

## 3.4. Metabolomics study

Reproducibility of the metabolomics approach was determined from six replicated analyses of the same quality control sample (QC) interspersed throughout the analysis. The relative standard deviation (RSD) of the peak area of all metabolites is below 20% (electronic supplemental material, figures S1 and S2), which demonstrates good stability and reproducibility [19]. S-plot analysis was employed by Simca v. 14.1 to determine the specific variation between control and model groups, while variable importance in the projection (VIP) values (VIP greater than 1) of variables were generated and identified using OPLS-DA analysis [20] among control, model and baicalin groups (figure 4). According to the above criteria, 35 metabolites involved in the AR model were selected, and 32 of these were identified based on retention time and accurate elemental compositions with the available databases, such as METLIN (http://metlin.scripps.edu), HMDB (http://www.hmdb.ca) and KEGG (http://genome.jp/kegg) (electronic supplemental material, table S1). The p-value was set to 0.05 when determining significantly different metabolites in this study. Next, 30 significantly different metabolites were selected for further study, and 21 metabolites were confirmed with available reference standards by matching their retention time and accurate elemental compositions.

As shown in table 1, the levels of arachidonic acid, palmitic acid, γ-linolenic acid, α-ketoglutaric acid, tetradecanedioic acid, glutamine, PC (20 : 4/0 : 0), indoleacrylic acid, betaine, alanine, serine and PC

**6**

**Table 1.** The information for 30 significantly different metabolites. Compared to control group, $*p < 0.05$, $**p < 0.01$; Compared to model group, $\#p < 0.05$, $\#\#p < 0.01$; '↑' means a higher level of metabolites, whereas '↓' represents a lower level of metabolites.

| no. | ion mode | r.t. (min) | m/z | VIP | elemental composition | metabolites | control group | model group[a] | baicalin group[b] |
|---|---|---|---|---|---|---|---|---|---|
| 1 | $[M-H]^-$ | 11.404 | 303.2386 | 12.75 | $C_{20}H_{32}O_2$ | arachidonic acid[c] | 747.39 ± 41.24 | 385.69 ± 21.83***↓ | 661.68 ± 30.81##↑ |
| 2 | $[M-H]^-$ | 11.337 | 255.2337 | 10.15 | $C_{16}H_{32}O_2$ | palmitic acid[c] | 902.98 ± 17.15 | 656.12 ± 66.22**↓ | 842.41 ± 44.67##↑ |
| 3 | $[M-H]^-$ | 11.202 | 279.2338 | 7.39 | $C_{18}H_{32}O_2$ | linoleic acid | 985.79 ± 48.63 | 1110.77 ± 77.21**↑ | 1009.47 ± 62.61##↓ |
| 4 | $[M-H]^-$ | 11.045 | 277.2166 | 5.73 | $C_{18}H_{30}O_2$ | γ-linolenic acid | 213.02 ± 29.11 | 131.22 ± 16.67**↓ | 193.95 ± 22.25##↑ |
| 5 | $[M-H]^-$ | 5.211 | 178.0521 | 4.03 | $C_9H_9NO_3$ | hippuric acid[c] | 15.58 ± 2.72 | 52.38 ± 8.72**↑ | 25.11 ± 3.01##↓ |
| 6 | $[M-H]^-$ | 1.128 | 89.0264 | 3.48 | $C_3H_6O_3$ | lactic acid[c] | 3.33 ± 0.23 | 31.5 ± 6.02**↑ | 10.13 ± 1.84##↓ |
| 7 | $[M-H]^-$ | 10.225 | 335.2249 | 2.48 | $C_{20}H_{32}O_4$ | LTB4[c] | 2.64 ± 0.3 | 16.61 ± 2.33**↑ | 6.03 ± 0.74##↓ |
| 8 | $[M-H]^-$ | 1.121 | 145.0139 | 2.39 | $C_5H_6O_5$ | α-ketoglutaric acid[c] | 14.95 ± 2.93 | 1.74 ± 0.31**↓ | 11.79 ± 2.16##↑ |
| 9 | $[M-H]^-$ | 1.124 | 191.0121 | 1.54 | $C_6H_8O_7$ | citric acid[c] | 5.24 ± 0.38 | 10.57 ± 1.59**↑ | 6.54 ± 0.55##↓ |
| 10 | $[M-H]^-$ | 4.711 | 181.0522 | 1.53 | $C_9H_{10}O_4$ | hydroxyphenyllactic acid | 1.77 ± 0.27 | 7.09 ± 0.89**↑ | 3.15 ± 0.21##↓ |
| 11 | $[M-H]^-$ | 9.874 | 257.1752 | 1.19 | $C_{14}H_{26}O_4$ | tetradecanedioic acid | 7.58 ± 1.17 | 4.09 ± 0.65**↓ | 6.82 ± 0.79##↑ |
| 12 | $[M+H]^+$ | 1.116 | 115.0099 | 9.34 | $C_4H_4O_4$ | fumarate[c] | 26.15 ± 3.16 | 68.32 ± 12.58**↑ | 37.15 ± 3.8##↓ |
| 13 | $[M+H]^+$ | 6.601 | 147.9719 | 6.51 | $C_4H_6NO$ | glutamine[c] | 96.71 ± 8.88 | 73.69 ± 3.51**↓ | 91.79 ± 6.43##↑ |
| 14 | $[M+H]^+$ | 1.106 | 133.0196 | 6.03 | $C_4H_6O_5$ | malate[c] | 9.38 ± 1.7 | 26.95 ± 3.03**↑ | 13.95 ± 1.01##↓ |
| 15 | $[M+H]^+$ | 10.717 | 544.3403 | 5.98 | $C_{28}H_{50}NO_7P$ | PC(20:4/0:0) | 117.21 ± 3.86 | 97.25 ± 6.23**↓ | 113.26 ± 3.84##↑ |
| 16 | $[M+H]^+$ | 7.601 | 170.0935 | 5.84 | $C_7H_{11}N_3O_2$ | 3-methylhistidine | 39.66 ± 4.71 | 58.02 ± 4.77**↑ | 44.71 ± 3.43##↓ |
| 17 | $[M+H]^+$ | 7.632 | 175.1203 | 5.03 | $C_6H_{14}N_4O_2$ | arginine[c] | 115.13 ± 10.11 | 133.07 ± 10.41**↑ | 120.81 ± 8.97##↓ |
| 18 | $[M+H]^+$ | 4.581 | 188.0719 | 4.25 | $C_{11}H_9NO_2$ | indoleacrylic acid | 82.11 ± 7.27 | 69.8 ± 5.43**↓ | 79.77 ± 5.82##↑ |
| 19 | $[M+H]^+$ | 5.610 | 182.0825 | 3.46 | $C_9H_{11}NO_3$ | tyrosine[c] | 18.12 ± 1.6 | 25.16 ± 3.34**↑ | 20.09 ± 1.34##↓ |
| 20 | $[M+H]^+$ | 10.590 | 317.2118 | 3.07 | $C_{20}H_{30}O_3$ | LTA4[c] | 0.28 ± 0.05 | 4.77 ± 0.89**↑ | 1.43 ± 0.21##↓ |
| 21 | $[M+H]^+$ | 0.912 | 114.0655 | 2.58 | $C_4H_7N_3O$ | creatinine[c] | 11.67 ± 1.21 | 15.19 ± 0.97**↑ | 12.68 ± 0.92##↓ |
| 22 | $[M+H]^+$ | 4.990 | 132.1023 | 2.16 | $C_6H_{13}NO_2$ | leucine[c] | 5.44 ± 0.96 | 2.79 ± 0.46**↓ | 4.81 ± 0.68##↑ |

(Continued.)

**Table 1.** (Continued.)

| no. | ion mode | r.t. (min) | m/z | VIP | elemental composition | metabolites | control group | model group[a] | baicalin group[b] |
|---|---|---|---|---|---|---|---|---|---|
| 23 | $[M+H]^+$ | 0.831 | 118.0859 | 1.95 | $C_5H_{11}NO_2$ | betaine[c] | 24.6 ± 2.98 | 21.13 ± 2.65**↓ | 23.95 ± 1.71#↑ |
| 24 | $[M+H]^+$ | 7.402 | 112.0888 | 1.78 | $C_5H_9N_3$ | histamine[c] | 0.17 ± 0.03 | 1.7 ± 0.29**↑ | 0.56 ± 0.08##↓ |
| 25 | $[M+H]^+$ | 4.202 | 154.0979 | 1.61 | $C_7H_{11}N_3O$ | N-acetylhistamine | 0.43 ± 0.05 | 1.71 ± 0.3**↑ | 0.76 ± 0.08##↓ |
| 26 | $[M+H]^+$ | 6.102 | 90.0560 | 1.58 | $C_3H_7NO_2$ | alanine[c] | 5.79 ± 0.66 | 4.27 ± 0.46**↓ | 5.46 ± 0.51##↑ |
| 27 | $[M+H]^+$ | 6.851 | 133.0610 | 1.58 | $C_4H_8N_2O_3$ | asparagine[c] | 13.67 ± 0.88 | 15.48 ± 1.44**↑ | 14.26 ± 0.83#↓ |
| 28 | $[M+H]^+$ | 6.801 | 106.0521 | 1.42 | $C_3H_7NO_3$ | serine[c] | 10.1 ± 0.79 | 8.65 ± 0.79**↓ | 9.83 ± 0.62##↑ |
| 29 | $[M+H]^+$ | 5.411 | 118.0871 | 1.01 | $C_5H_{11}NO_2$ | valine[c] | 1.36 ± 0.13 | 1.92 ± 0.26**↑ | 1.52 ± 0.09##↓ |
| 30 | $[M+H]^+$ | 10.882 | 496.3403 | 3.71 | $C_{24}H_{50}NO_7P$ | PC (16 : 0/0 : 0) | 166 ± 4.3 | 156.03 ± 6.9**↓ | 165.07 ± 2.75##↑ |

[a]Compared to control group, $*p < 0.05$, $**p < 0.01$.
[b]Compared to model group, $\#p < 0.05$, $\#\#p < 0.01$.
[c]Confirmed with authentic standards.

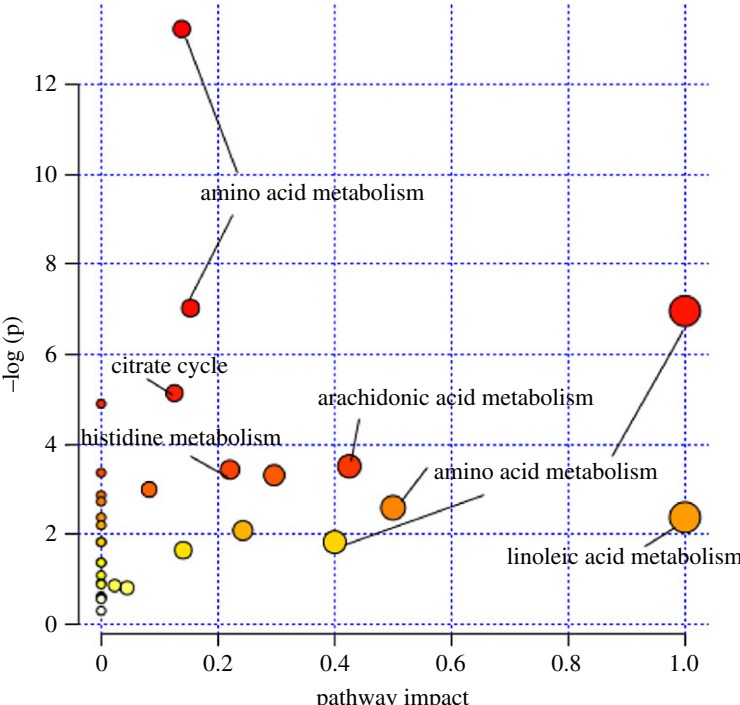

**Figure 5.** The metabolic pathway of the altered metabolites analysed by MetaboAnalyst 4.0.

$(16:0/0:0)$ were observed to be significantly decreased in the model group $(p < 0.05)$. The levels of linoleic acid, hippuric acid, lactic acid, LTB4, citric acid, hydroxyphenyllactic acid, fumarate, malate, 3-methylhistidine, arginine, tyrosine, histamine, N-acetylhistamine, asparagine, valine, LTA4 and creatinine were observed to be markedly increased in the model group $(p < 0.05)$. In the baicalin-treated group, the apparent trend of these metabolite levels was the opposite of the model group.

MetaboAnalyst 4.0 (http://www.metaboanalyst.ca) was used to analyse metabolic pathways and showed that several pathways, such as arachidonic acid metabolism, citrate cycle, histidine metabolism, linoleic acid metabolism and other amino metabolism, were disordered in the AR model group or in the baicalin group (figure 5).

Several altered metabolites are of special interest because they are directly related to allergic diseases. For instance, linoleic acid (VIP = 7.39), which can induce red blood cells and haemoglobin damage via oxidative mechanisms, is a critical component of polyunsaturated fatty acids [21] and can be converted into γ-linolenic acid (VIP = 5.73) and arachidonic acid (VIP = 12.75). The decreased level of γ-linolenic acid and elevated levels of linoleic acid and arachidonic acid in the model group indicated that linoleic acid metabolism was disordered, and this disorder can be changed by baicalin treatment. Arachidonic acid can generate LTA4 (VIP = 3.07) and LTB4 (VIP = 2.48) via the 5-lipid oxidase pathway. In this study, we observed that LTA4 and LTB4 were increased, followed by arachidonic acid, which is in accordance with this theory [22]. LTB4, synthesized by LTA4 hydrolase from LTA4, is a leukotriene involved in inflammation that plays an important role in the development of AR [23]. These results indicated that arachidonic acid metabolism was also disordered in our study, and with high VIP, arachidonic acid might be used as a specific candidate biomarker for AR. Histamine (VIP = 1.78) was detected by ELISA and was also identified and quantified by UPLC–Q-TOF–MS/MS. Moreover, N-acetylhistamine (VIP = 1.61) and 3-methylhistidine (VIP = 5.84) generated from histamine were also identified and quantified. Interestingly, histamine was described as the 'gold standard' for allergic reactions [24], but in this study, the VIP value of histamine was not specific, which may be related to the short half-life of histamine [18]. By contrast, 3-methylhistidine with high VIP may be a new indicator for early-phase allergenic mediators.

Furthermore, a large number of amino acids were significantly altered in AR. Our results suggested that rats sensitized by OVA may consume more energy in order to fight AR when compared to normal rats. Abnormal changes, such as fumarate, malate and α-ketoglutaric acid, belonging to the citric acid cycle, confirm this view. The citric acid cycle, as the linkage among sugar, lipids and amino acid metabolism, is the most important energy source for the body [25]. In addition, other amino acids are

substrates for metabolic energy, which could be used for the synthesis of specific proteins, peptides and other nitrogenous compounds [26]. In addition, the changes in PC (20 : 4/0 : 0) and PC (16 : 0/0 : 0) levels, as the precursor of lysoPC, which has been confirmed to be a chemo-attractant for T lymphocytes, showed that the OVA-sensitized rats are in an inflammation state [27].

A similar study of AR treated by baicalin was recently published by another group using guinea pigs. The study focused on IgE, histamine and inflammatory cytokines, and the results suggested that baicalin may be a useful drug in the treatment of AR, which was the same as our results. Additionally, the index IL-8 was replaced with IL-4 in this study, and a randomized double-blind trial with AR showed that IL-4 is important in regulating IgE-mediated allergy [28,29]. IL-4 could also significantly amplify itching symptoms such as histamine [30]; therefore, IL-4 is more closely associated with AR. In addition, the metabolic mechanism of AR and the systematic mechanism of baicalin against OVA-sensitized AR were explored in this study. Other allergy mediators that are specific to AR should be detected and identified, and the different time period mechanisms of baicalin on AR need to be further studied.

## 4. Conclusion

Our study indicated that baicalin has a protective effect on rats with AR by inhibiting the release of inflammatory mediators. Furthermore, metabolomics was applied to systematically study the AR induced by OVA and the mechanism of baicalin on AR. We conclude that baicalin is anti-AR not only because it suppressed the production of inflammatory mediators but also because it regulated arachidonic acid metabolism, the citric acid cycle, histidine metabolism, linoleic acid metabolism and other amino acid metabolism.

Ethics. The animals were maintained under SPF laboratory conditions and received human care according to the guidelines of the Local Institutes of Health guide for the care and use of laboratory animals.
Data accessibility. The datasets supporting this article have been uploaded as part of the electronic supplementary material.
Authors' contributions. S.C., G.C. and Y.X. had substantial contributions to the conception and design, analysis and interpretation of data; S.S. and X.M. contributed to the acquisition of data, analysis and interpretation of data; S.C., G.C. and Y.X. drafted the article or revised it critically for important intellectual content; Y.X. made the final approval of the version to be published; Y.X. agreed to be accountable for all aspects of the work in ensuring that questions related to the accuracy or integrity of any part of the work are appropriately investigated and resolved.
Competing interests. We have no competing interests.
Funding. These studies were supported by funding obtained from the Natural Science Foundation of China (grant no. 81303205), Shenyang Young and Middle-aged Science and Technology Innovation Talents Support Program (RC170344), Shenyang Science and Technology Plan Project (18-013-78), Liaoning University Innovation Talent Support Program (LR2017002).
Acknowledgements. Beijing University of Chemical Technology contributed to the study.

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
