## [Reviewer comments · Royal Society Open Science]

Review History

RSOS-181081.R0 (Original submission)

Review form: Reviewer 1

Is the manuscript scientifically sound in its present form?

Yes

Are the interpretations and conclusions justified by the results?

Yes

Is the language acceptable?

Yes

Is it clear how to access all supporting data?

Yes

Do you have any ethical concerns with this paper?

No

Have you any concerns about statistical analyses in this paper?

No

Recommendation?

Accept with minor revision (please list in comments)

Comments to the Author(s)

1. Acronyms should be better defined in the abstract and introduction. E.g. IL-4, SD.
2. S-plot and OPLS-DA analysis were used in this study, the software package Simca version 14.1 (Umetrics, Sweden) was used to process OPLS-DA analysis, author should defined which software was used to process S-plot analysis, Masslynx or Simca version 14.1.
3. Author had demonstrated "Reproducibility of the metabolomics was determined from six replicated analyses of the same quality control sample (QC) interspersed throughout the analysis. And the RSD of the peak area of all metabolites are below 20%, which demonstrated good stability and reproducibility" at the part of 4.4 Metabolomics study, the figure of PCA analysis(including QC data) should be provided in the supplemental materials.

Review form: Reviewer 2

Is the manuscript scientifically sound in its present form?

Yes

Are the interpretations and conclusions justified by the results?

Yes

Is the language acceptable?

Yes

Is it clear how to access all supporting data?

Yes

Do you have any ethical concerns with this paper?

No

Have you any concerns about statistical analyses in this paper?

No

Recommendation?

Accept with minor revision (please list in comments)

Comments to the Author(s)

The research part is well-organized, with clear and detailed step-by-step methods. However, the language quality is not satisfactory and required to be improved. My remarks/queries to the authors are as the attached file (Appendix A).

Decision letter (RSOS-181081.R0)

28-Sep-2018

Dear Dr Xu,

The editors assigned to your paper ("Metabolomics analysis of baicalin on ovalbumin sensitized allergic rhinitis rats") have now received comments from reviewers. We would like you to revise your paper in accordance with the referee and Associate Editor suggestions which can be found below (not including confidential reports to the Editor). Please note this decision does not guarantee eventual acceptance.

Please submit a copy of your revised paper before 21-Oct-2018. Please note that the revision deadline will expire at 00.00am on this date. If we do not hear from you within this time then it will be assumed that the paper has been withdrawn. In exceptional circumstances, extensions may be possible if agreed with the Editorial Office in advance. We do not allow multiple rounds of revision so we urge you to make every effort to fully address all of the comments at this stage. If deemed necessary by the Editors, your manuscript will be sent back to one or more of the original reviewers for assessment. If the original reviewers are not available, we may invite new reviewers.

- Data accessibility

If you wish to submit your supporting data or code to Dryad (<http://datadryad.org/>), or modify your current submission to dryad, please use the following link:
<http://datadryad.org/submit?journalID=RSOS&manu=RSOS-181081>

- **Competing interests**

- **Authors' contributions**

- **Acknowledgements**

- **Funding statement**

Please note that Royal Society Open Science charge article processing charges for all new submissions that are accepted for publication. Charges will also apply to papers transferred to Royal Society Open Science from other Royal Society Publishing journals, as well as papers submitted as part of our collaboration with the Royal Society of Chemistry (<http://rsos.royalsocietypublishing.org/chemistry>). If your manuscript is newly submitted and subsequently accepted for publication, you will be asked to pay the article processing charge, unless you request a waiver and this is approved by Royal Society Publishing. You can find out more about the charges at <http://rsos.royalsocietypublishing.org/page/charges>. Should you have any queries, please contact openscience@royalsociety.org.

Kind regards,

Andrew Dunn

on behalf of Professor Savvas Savvides (Associate Editor) and Kevin Padian (Subject Editor)
openscience@royalsociety.org

Associate Editor's comments:

Please ensure that you fully respond to the referee commentary, in particular their observation that the quality of the writing needs to be improved. While we sympathise with the complexities and idiosyncrasy of the English language, you may benefit from the advice of services such as these: <https://royalsociety.org/journals/authors/language-polishing/>.

Editor comments:

Thanks for your submission. Please make sure that this manuscript is fully edited by an expert who is a native speaker of English (a very difficult language! sorry). We can only publish the paper if the language is up to the standards of the journal. Best of luck.

Comments to Author:

Reviewers' Comments to Author:

Reviewer: 1

Comments to the Author(s)

1. Acronyms should be better defined in the abstract and introduction. E.g. IL-4, SD.
2. S-plot and OPLS-DA analysis were used in this study, the software package Simca version 14.1 (Umetrics, Sweden) was used to process OPLS-DA analysis, author should defined which software was used to process S-plot analysis, Masslynx or Simca version 14.1.
3. Author had demonstrated "Reproducibility of the metabolomics was determined from six replicated analyses of the same quality control sample (QC) interspersed throughout the analysis. And the RSD of the peak area of all metabolites are below 20%, which demonstrated good stability and reproducibility" at the part of 4.4 Metabolomics study, the figure of PCA analysis(including QC data) should be provided in the supplemental materials.

Reviewer: 2

Comments to the Author(s)

The research part is well-organized, with clear and detailed step-by-step methods. However, the language quality is not satisfactory and required to be improved. My remarks/queries to the authors are as the attached file.

Author's Response to Decision Letter for (RSOS-181081.R0)

See Appendix B.

RSOS-181081.R1 (Revision)

Review form: Reviewer 1

Is the manuscript scientifically sound in its present form?

Yes

Are the interpretations and conclusions justified by the results?

Yes

Is the language acceptable?

Yes

Is it clear how to access all supporting data?

Yes

Do you have any ethical concerns with this paper?

No

Have you any concerns about statistical analyses in this paper?

No

Recommendation?

Accept as is

Comments to the Author(s)

Thanks for your contribution to the research on AR, i learned a lot from your painstaking work. Your manuscript was very meaningful with clear points and conclusion, it was well organised and stated. Be patient and keep moving, wish you can make great achievements in future research.

Review form: Reviewer 2

Is the manuscript scientifically sound in its present form?

Yes

Are the interpretations and conclusions justified by the results?

Yes

Is the language acceptable?

Yes

Is it clear how to access all supporting data?

Yes

Do you have any ethical concerns with this paper?

No

Have you any concerns about statistical analyses in this paper?

No

Recommendation?

Accept as is

Comments to the Author(s)

The author has revised it according to my comments.

Decision letter (RSOS-181081.R1)

25-Jan-2019

Dear Dr Xu,

I am pleased to inform you that your manuscript entitled "Metabolomics analysis of baicalin on ovalbumin sensitized allergic rhinitis rats" is now accepted for publication in Royal Society Open Science.

Kind regards,
Andrew Dunn
Senior Publishing Editor
Royal Society Open Science
openscience@royalsociety.org

on behalf of Professor Savvas Savvides (Associate Editor) and Kevin Padian (Subject Editor)
openscience@royalsociety.org

Reviewer comments to Author:

Reviewer: 2

Comments to the Author(s)

The author has revised it according to my comments.

Reviewer: 1

Comments to the Author(s)

Thanks for your contribution to the research on AR, i learned a lot from your painstaking work. Your manuscript was very meaningful with clear points and conclusion, it was well organised and stated. Be patient and keep moving, wish you can make great achievements in future research.

福昕PDF编辑器

· 永久 · 轻巧 · 自由

升级会员

批量购买

永久使用

无限制使用次数

极速轻巧

超低资源占用，告别卡顿慢

自由编辑

享受Word一样的编辑自由

扫一扫，关注公众号

**ROYAL SOCIETY
OPEN SCIENCE****Metabolomics analysis of baicalin on ovalbumin sensitized
allergic rhinitis rats**

Journal:	Royal Society Open Science
Manuscript ID	RSOS-181081
Article Type:	Research
Date Submitted by the Author:	03-Jul-2018
Complete List of Authors:	Chen, Saizhen; a. Taizhou Central Hospital (Taizhou University Hospital) Chen, Guirong; Liaoning University of Traditional Chinese Medicine - Dalian Campus Shu, Sheng; a. Taizhou Central Hospital (Taizhou University Hospital) Xu, Yubin; a. Taizhou Central Hospital (Taizhou University Hospital) Ma, Xiande; Liaoning University of Traditional Chinese Medicine
Subject:	biochemistry < BIOLOGY, Analytical chemistry < CHEMISTRY, Biochemistry < CHEMISTRY
Keywords:	metabolomics, baicalin, allergic rhinitis
Subject Category:	Biology (whole organism)

Metabolomics analysis of baicalin on ovalbumin sensitized allergic rhinitis rats

Saizhen Chen^{a, #}, Guirong Chen^{b, #}, Sheng Shu^a, Yubin Xu^{a, *}, Xiande Ma^b

^aTaizhou Central Hospital (Taizhou University Hospital), Taizhou, 318000, China.

^bLiaoning University of Traditional Chinese Medicine, Shenyang, 110847, China.

Keywords: metabolomics; baicalin; allergic rhinitis

1. Summary revise

Allergic rhinitis (AR) is a global health problem, appears in all age groups and affects nearly 15% to 30% people. Baicalin is been used for the treatment of various allergic diseases including AR. However, the metabolic mechanisms of AR and baicalin against AR were not systematically studied. Here, ovalbumin-sensitized AR rats were used as model, animal behavior, histological analysis, ELISA method and metabolomics were used to elucidate the mechanism of baicalin for AR. The result indicated that baicalin has a protective effect on allergic rhinitis rats via inhibiting the release of IgE, histamine, IL-1 β , IL-4, IL-6 and TNF- α . In addition, ovalbumin induced AR include arachidonic acid, LTA4, LTB4, α -ketoglutaric acid, PC(20:4/0:0), PC(16:0/0:0), citric acid, fumarate, malate, 3-methylhistidine, histamine and other amino acids were confirmed to be involved in arachidonic acid, histidine metabolism, TCA cycle and amino acid metabolism, and these disorders could be alleviated or reversed by baicalin.

2. Introduction

Allergic rhinitis (AR) is a type of chronic inflammation in the respiratory tract [1]. It is a very common disease of nasal mucosa, triggered by the interaction of specific immunoglobulin E (IgE) and ubiquitous environmental proteins in sensitized patients [2]. The main symptoms of AR are sneezing, rhinorrhea, itching and nasal congestion; these symptoms are usually accompanied by other related symptoms such as nasal mucosa swelling [3, 4]. Nowadays, AR appears in all age groups, affect large part of the global population, is a major public health burden [5]. Epidemiologic data showed that AR affects nearly 15% to 30% people [6, 7].

Current studies have predominantly concentrated on cytokines, chemokines such as IL-1 β , TNF- α and IL-4 involved in pathogenesis and process of allergic reactions during AR, but the effective metabolites related to AR still unclear [8, 9]. It is imperative to explore potential metabolites to systematically indicate the mechanism of AR.

Recently, the use of natural products for treating AR has been increasing worldwide; many prior trials have evaluated the effectiveness of herbal medicines for AR [10]. Baicalin, a flavonoid compound isolated from *Scutellaria baicalensis* Georgi, has been shown to have multiple pharmacological activities including anti-allergic, anti-inflammatory and antioxidant activities. Based on its anti-allergic and anti-inflammatory activity, baicalin is widely used to treat various allergic, inflammatory diseases including asthma, atopic dermatitis and AR [11]. However, there is little information about the metabolic mechanism of baicalin on Ovalbumin (OVA)-induced allergic rhinitis SD rats.

In this study, untargeted metabolomics was used to evaluate the metabolic mechanism of allergic rhinitis SD rats, based on this, systematical mechanism of baicalin against OVA-induced AR was explored. This study might provide new insights to understand AR as well as the potential application of baicalin as a drug for AR treatment.

3. Materials and Methods

3.1 Reagents

Baicalin was purchased from National Institute for the control of Food and Drug (Beijing, China). HPLC-grade methanol, acetonitrile and formic acid were purchased from Merck (Germany). The ELISA kits for IgE, histamine, IL-4, IL-1 β , IL-6 and TNF- α were purchased from Usen Life Science Inc. (Wuhan, China). OVA, leucine-enkephalin and all the authentic standards were purchased from Sigma-Aldrich (St. Louis, MO, USA).

*Author for correspondence (xuyubin1988@126.com).

†Present address: Taizhou Central Hospital (Taizhou University Hospital), Taizhou, 318000, China

#These authors contributed equally to this work.

3.2 Animal experiments and sample collection

Male SD rats (200±20 g) were obtained from Liaoning Changsheng Biotechnology Co. Ltd. (Liaoning, China). The animals were maintained under SPF laboratory conditions and received human care according to the guidelines of the Local Institutes of Health guide for the care and use of laboratory animals.

All SD rats were randomly divided into three groups, with 10 rats in each group: control group, model group and baicalin group. First, the model group and baicalin group were sensitized by OVA by using a standard protocol as previous study [12]. Briefly, the rats were sensitized by OVA via injection on days 1, 5, and 10; from days 15 to 21, the sensitized rats were intranasal challenged by daily droppings with OVA. After the AR models successfully established, the rats of baicalin group were orally administered baicalin for 10 days from days 22 to 31 at dose of 200 mg/kg, which was determined by the reference [2]; while the rats of model group were received saline with the same days. The control group rats were received saline alone on the same schedule.

The numbers of sneezes and nose scratching were counted for 15 minutes after 2 h of the last administration. Then the rats were sacrificed by drawing out all the blood from abdominal aorta to obtain blood samples after isoflurane anesthesia and the nasal mucosa was separated and fixed in 4% paraformaldehyde at room temperature. The blood samples were centrifuged at 3000×g for 10 min at 4°C to obtain serum, which was then stored at -80°C until analysis.

3.3 Histological analysis

The nasal tissues were dehydrated, embedded in paraffin, sectioned at 4 μm thickness and stained with hematoxylin and eosin, then analyzed with light microscopy.

3.4 Measurement of IgE, histamine, IL-1β, IL-4, IL-6 and TNF-α level of the serum

IgE, histamine, IL-1β, IL-4, IL-6 and TNF-α level of the serum were determined using ELISA kits according to the manufacturer's instructions.

3.5 Metabolomics

3.5.1 Sample preparation and UPLC/Q-TOF-MS/MS analysis. The method was established as previously described with minor modification [13]. 200μL Serum was added into 600μL acetonitrile, vortex-mixed for 30s and centrifuged at 12,000×g for 5 min to precipitate the proteins. Next, 600μL supernatant was collected and dried with vacuum drying centrifuged at room temperature. Then the dried residue was reconstituted with 100μL of acetonitrile-water (1:1, v/v). After centrifugation at 12,000×g for 5 min, an aliquot of 3μL was injected for UPLC/Q-TOF-MS/MS analysis.

A Waters Acquity UPLC system was used to perform the process. The chromatographic column was HSS T3 column (2.1 mm × 100mm, 1.7 μm, waters, USA) and the column temperature was maintained at 40 °C. The mobile phase was: methanol (A) and 0.1% formic acid (B), gradient elution condition was described as follows: 0 → 7 min: 0% → 60%A; 7 → 10 min: 60% → 100%A; 10 → 14 min: 100%A; 14 → 16 min: 100% → 0%A; Curve: 6.

Mass spectrometry was performed on a Waters Xevo G2 QTOF quadrupole accelerated time of flight mass spectrometer in positive and negative ion mode with an electrospray ionization (ESI) source. The desolvation gas was 800 L/hr, the desolvation temperature was 400°C, the cone gas was 50 L/hr and the temperature of ion source was 110°C; Capillary voltage, cone voltage, scanning time and collision energy were set as 3.0 kV, 45 V, 1 s and 6 eV, respectively. The spectra were collected every 0.2 s alternately with the interval at 0.02 s, mass range was from 50 to 1200Da. The m/z 556.2771([M+H]) and 554.2615[M-H] was for LockSpray™ leucine-enkephalin.

3.5.2 Data process. The data process was performed as previous. Briefly, the raw MS spectra were analyzed using MarkerLynx version 4.1 (Waters Corp., UK). Then the processed data list was exported and processed by orthogonal partial least squares discriminant analysis (OPLS-DA) using the software package Simca version 14.1 (Umetrics, Sweden).

3.6 Statistical analysis

All values were expressed as mean±SD. The significance of differences among the means of control, model and baicalin groups was compared through one-way Anova test using the Statistical Package for Social Sciences program (SPSS 20.0, USA). The significance threshold was set at p<0.05 for this test.

4. Results and Discussion

4.1 Animal behavior study

The nasal symptoms of rats were recorded for 15 min. The numbers of nasal sneezes and scratching were significantly increased in model group, while significantly decreased in baicalin group. Scores for groups are shown in Figure 1, the results indicated that the OVA-sensitized rat model was successfully established, and the symptoms can be inhibited by baicalin at dose of 200 mg/kg.

4.2 Histological analysis

Histological analysis was used to evaluate the pathological changes of nasal mucosa of rats in each group. The nasal mucosa was arranged neatly and had no abnormality in control group, while a large number of eosinophilic granulocytes infiltrated, and blood stasis was obvious observed in model group, the pathological changes were significantly alleviated by baicalin (Figure 2).

4.3 Effects of baicalin on IgE, histamine, IL-1β, IL-4, IL-6 and TNF-α level in serum

To determine the effect of baicalin on AR models, the level of IgE, histamine, IL-1β, IL-4, IL-6 and TNF-α were measured by ELISA. The level of IgE, histamine, IL-1β, IL-4, IL-6 and TNF-α in model group were markedly increased than control group; and 200 mg/kg baicalin were significantly decreased the level of IgE, histamine, IL-1β, IL-4, IL-6 and TNF-α in OVA-sensitized rats. (Figure 3)

AR is caused by an IgE-mediated inflammatory reaction, including early- and late-phase allergenic mediators [14]. Histamine, as the most iconic indicator of vascular vitality in allergic reactions, is one of the early-phase allergenic mediators; however, histamine as a detection index still has some limitations, mainly reflected in the short half-life [15,

16]. The late-phase allergenic mediators include IL-1 β , IL-4, IL-6, the proinflammatory cytokine TNF- α , which can induce IgE synthesis, play an essential role in AR development [17, 18].

4.4 Metabolomics study

Reproducibility of the metabolomics was determined from six replicated analyses of the same quality control sample (QC) interspersed throughout the analysis. And the RSD of the peak area of all metabolites are \leq 20%, which demonstrated good stability and reproducibility [19]. S-plot analysis was employed to determine the specific variation between control and model groups, while variable importance in the projection (VIP) values (VIP>1) of variables were generated and identified using OPLS-DA analysis [20] among control, model and baicalin groups (Figure 4). According to the above criteria, 35 significantly metabolites related to AR were selected and 32 of these were identified on the basis of retention time and accurate elemental compositions with the available databases such as METLIN(<http://metlin.scripps.edu>), HMDB(<http://www.hmdb.ca>), KEGG(<http://genome.jp/kegg>) (Supplemental Table 1). The p-value was set to 0.05 for significantly differential metabolites in this study. Following this, 30 significantly differential metabolites were selected for further study and 21 metabolites were confirmed with available reference standards by matching their retention time and accurate elemental compositions.

As shown in Table 1, the levels of arachidonic acid, palmitic acid acid, linolenic acid, α -ketoglutaric acid, tetradecanedioic acid, glutamine, PC (20:4/0:0), indoleacrylic acid, betaine, alanine, serine and PC (16:0/0:0) were observed to be significantly decreased in model group ($p < 0.05$); The levels of linoleic acid, hippuric acid, lactic acid, LTB₄, citric acid, hydroxyphenyllactic acid, fumarate, malate, 3-methylhistidine, arginine, tyrosine, histamine, N-acetylhistamine, asparagine, valine, LTA₄, creatinine were observed to be markedly increased in model group ($p < 0.05$), while administrated with baicalin, the apparent trend of these metabolites is opposite to that of the model group.

MetaboAnalyst 4.0 (<http://www.metaboanalyst.ca>) was used to analysis metabolic pathway, which showed that several pathways such as arachidonic acid metabolism, citrate cycle, histidine metabolism, linoleic acid metabolism and other amino metabolism were disordered when rats in AR status or AR rats administrated with baicalin. (Figure 5)

Several altered metabolites are of special interest since they are directly related to allergic diseases. For instance, linoleic acid (VIP=7.39), which can induces red blood cells and hemoglobin damage via oxidative mechanism, is a critical component of polyunsaturated fatty acids [21], can be converted into γ -linolenic acid (VIP=5.73) and arachidonic acid (VIP=12.75). The declined level of γ -linolenic acid and elevated levels of linoleic acid and arachidonic acid in model group indicated the linoleic acid metabolism was disordered, while this disorder can be changed by baicalin. Arachidonic acid can generate LTA₄ (VIP=3.07) and LTB₄ (VIP=2.48) via 5-lipid oxidase pathway, then, LTA₄ and LTB₄ were increased followed by arachidonic acid in this study which was in accordance with this theory [22]. LTB₄, synthesized by LTA₄ hydrolase from LTA₄ is a leukotriene involved in inflammation, playing an important role in the development of AR [23], these indicated that the arachidonic acid metabolism was also disordered in our study, and with the high VIP, arachidonic acid might be used as specific candidate biomarker for AR. Histamine (VIP=1.78) was detected by ELISA method but also identified and determined by UPLC-Q-TOF-MS/MS, moreover, N-acetylhistamine (VIP=1.61) and 3-methylhistidine (VIP=5.84) generated from histamine were also identified and determined. To our interesting, histamine was described as "gold standard" for allergic reaction[24], but in this study, the VIP value of histamine was not specific, which may be related to the short half-life of histamine[18]; on the contrary, 3-methylhistidine with high VIP may be able to regard as new indicator for the early-phase allergenic mediators.

Furthermore, a large number of amino acids were significantly altered during AR status. Our result suggested that rats sensitized by OVA may consume a lot of energy to fight with AR than normal rats, the abnormal changes such as fumarate, malate and α -ketoglutaric acid belonging to citrate cycle confirm this view. Citric cycle, as the linkage among sugar, lipids and amino acid metabolism, is the most important energy source for body [25]. Besides, other amino acids are substrates for metabolic energy, which could be used for specific protein, peptide and other nitrogenous compounds synthesis [26]. In addition, the abnormal changes of PC (20:4/0:0) and PC (16:0/0:0), as the precursor of LysoPC, which has been confirmed to be a chemo-attractant for T lymphocytes showed that the OVA-sensitized rats are in inflammation state [27].

A similar study about AR treated by baicalin had recently published by another group using guinea pigs. The study focused on the IgE, histamine and inflammatory cytokines, and results suggested that baicalin may be a useful drug in the treatment of AR, was the same as our results. Additionally, the index IL-8 was replaced with IL-4 in this study; a randomized double-blind trial with allergic rhinitis showed that IL-4 is important in regulating IgE-mediated allergy[28, 29]; IL-4 could also significantly amplify itching symptoms such as histamine[30], therefore, IL-4 is more closely associated with AR. Besides, the metabolic mechanism of AR and the systematical mechanism of baicalin against OVA-sensitized AR were explored in this study, other allergy mediators which were specific to allergic rhinitis should be detected and identified, the different time periods mechanism of baicalin on allergic rhinitis needs to be further studied.

5. Conclusion

Our studies indicated that baicalin has protective effect on rats with AR by inhibiting the release of inflammatory mediators. Furthermore, metabolomics were applied to study the AR induced by OVA and the mechanism of baicalin on AR systematically. We concluded baicalin anti-AR not only suppressed the productions of inflammatory mediators, but also regulated arachidonic acid metabolism, citrate cycle, histidine metabolism, linoleic acid metabolism and other amino acid metabolism.

Acknowledgments

Beijing University of Chemical Technology contributed to the study.

Ethical Statement

The animals were maintained under SPF laboratory conditions and received human care according to the guidelines of the Local Institutes of Health guide for the care and use of laboratory animals.

Funding Statement

These studies were supported by funding obtained from the Natural Science Foundation of China (No.81303205), Shenyang Young and Middle-aged Science and Technology Innovation talents support Program (RC170344), Shenyang Science and Technology Plan Project (18-013-78), Liaoning University Innovation Talent support Program (LR2017002).

Data Accessibility

The datasets supporting this article have been uploaded as part of the Supplementary Material.

Competing Interests

We have no competing interests.

Authors' Contributions

Saizhen Chen, Guirong Chen and Yubin Xu had substantial contributions to conception and design, analysis and interpretation of data; Sheng Shu and Xiande Ma had contributions to acquisition of data, analysis and interpretation of data; Saizhen Chen, Guirong Chen and Yubin Xu drafting the article or revising it critically for important intellectual content; Yubin Xu final approval of the version to be published; Yubin Xu agreed to be accountable for all aspects of the work in ensuring that questions related to the accuracy or integrity of any part of the work are appropriately investigated and resolved.

References

- K. Yamauchi, G. Tamura, T. Akasaka, T. Chiba, K. Honda, M. Kishi, H. Kobayashi, T. Kuronuma, A. Matsubara, T. Morikawa, H. Ogawa, N. Ohta, M. Okada, M. Sasaki, J. Saito, K. Sano, M. Satoh, Y. Shibata, Y. Takahashi, S. Takanashi, H. Inoue. 2009. *Allergol Int*, **58**, 55-61. (doi: 10.2332/allergolint.08-OA-0004)
- Y.J. Zhou, H. Wang, H.H. Sui, L. Li, C.L. Zhou, J.J. Huang. 2016. *Inflamm Res*, **65**, 603-612. (doi: 10.1007/s00011-016-0943-0)
- H.A. Kakli, T.D. Riley. 2016. *Prim Care*, **43**, 465-475. (doi: 10.1016/j.pop.2016.04.0094)
- J.A. Lee, S. Jang, J.H. Jun, M.S. Lee, E. Lee, N. Kim, D.H. Lee. 2018. *Medicine (Baltimore)*, **97**, e9551. (doi: 10.1097/MD.00000000000009551)
- D. Strachan, B. Sibbald, S. Weiland, N. Ait-Khaled, G. Anabwani, H.R. Anderson, M.I. Asher, R. Beasley, B. Björkstén, M. Burr, T. Clayton, J. Crane, P. Ellwood, U. Keil, C. Lai, J. Mallol, F. Martinez, E. Mitchell, S. Montefort, N. Pearce, C. Robertson, J. Shah, A. Stewart, M.E. Von, H. Williams. 1997. *Pediatr Allergy Immunol*, **8**, 161-176.
- R.A. Settipane, *Allergy Asthma Proc*, 2001, **22**, 185-189.
- A. Licari, G. Ciprandi, A. Marseglia, R. Castagnoli, S. Barberi, S. Caimmi, G.L. Marseglia. 2014. *Expert Rev Clin Immunol*, **10**, 1337-1347. (doi: 10.1586/1744666X.2014.955476)
- B.F. Marple. 2010. *Am J Rhinol Allergy*, **24**, 249-254. (doi: DOI: 10.2500/ajra.2010.24.3499)
- A.O. Eifan, S.R. Durham. 2016. *Clin Exp Allergy*, **46**, 1139-1151. (doi: 10.1111/cea.12780)
- C.C. Xue, F.C. Thien, J.J. Zhang, C.C. Da. C.G. Li. 2003. *Altern Ther Health Med*, **9**, 80-87.
- L. Xu, J. Li, Y. Zhang, P. Zhao, X. Zhang. 2017. *J Ethnopharmacol*, **208**, 199-206. (doi: 10.1016/j.jep.2017.07.013)
- C. Li, Y. Fu, Y. Wang, Y. Kong, M. Li, D. Ma, W. Zhai, H. Wang, Y. Lin, S. Liu, F. Ren, J. Li, Y. Wang. 2017. *Cell Biochem Funct*, **35**, 420-425. (doi: 10.1002/cbf.3291)
- Y. Xu, D. Dou, X. Ran, C. Liu, J. Chen. 2015. *J Chromatogr A*, **14**, 103-111. (doi: 10.1016/j.chroma.2015.09.019)
- C. Lv, Y. Zhang, L. Shen. 2018. *Int Arch Allergy Immunol*, **175**, 231-236. (doi: 10.1159/000486959)
- T. Hashimoto, H. Ohata, K. Honda. 2006. *J Pharmacol Sci*, **100**, 82-87.
- T. Kun, L. Jakubowski. 2012. *Pol J Radiol*, **77**, 19-24.
- Y. Shen, X. Ke, L. Yun, G.H. Hu, H.Y. Kang, S.L. Hong. 2018. *Mol Med Rep*, **17**, 1333-1339. (doi: 10.1155/2014/746846)
- L.A. Youssef, M. Schuyler, B.S. Wilson, J.M. Oliver. 2010. *Open Allergy J*, **3**, 91-101. (doi: 10.2174/1874838401003010091)
- L. Cui, H. Lu, Y.H. Lee. 2018. *Mass Spectrom Rev* (doi: 10.1002/mas.21562)
- Q.J. Yang, J.R. Zhao, J. Hao, B. Li, Y. Huo, Y.L. Han, L.L. Wan, J. Li, J. Huang, J. Lu, G.J. Yang, C. Guo. 2018. *J Cachexia Sarcopenia Muscle*, **9**, 71-85. (doi: 10.1002/jcsm.12246)
- T. Yuan, W.B. Fan, Y. Cong, H.D. Xu, C.J. Li, J. Meng, N.R. Bao, J.N. Zhao. 2015. *Int J Clin Exp Pathol*, **8**, 5044-5052.
- C.H. Hsu, C.M. Hu, K.H. Lu, S.F. Yang, C.H. Tsai, C.L. Ko, H.L. Sun, K.H. Lue. 2012. *Pediatr Neonatol*, **53**, 235-244. (doi: 10.1016/j.pedneo.2012.06.004)
- Q. Qiu, M. Xu, C. Lu, J. Chen, S. Chen, W. Kong, H. Han. 2016. *Int J Immunopathol Pharmacol*, **29**, 720-725. (doi: 10.1177/0394632016659301)
- Y. Xu, T. Kang, D. Dou, H. Kuang. 2015. *Asian Pac J Allergy Immunol*, **33**, 330-338. (doi: 10.12932/AP0619.33.4.2015)
- H. Song, W.L. Li, B.M. Liu, X.M. Sun, J.X. Ding, N. Chen. 2017. *RSC Adv*, **7**, 39403-39410. (doi: 10.1039/C7RA06930H)
- M.A. Grillo, A. Lanza, S. Colombatto. 2008. *Amino Acids*, **34**, 517-523. (doi: 10.1007/s00726-007-0006-5)
- X. Lu, X. Lian, J. Zheng, N. Ai, C. Ji, C. Hao. 2016. *RSC Adv*, **6**, 19545-19554. (doi: 10.1039/C5RA24301G)
- K. Appel, E. Munoz, C. Navarrete, C. Cruz-Teno, A. Biller, E. Thiemann. 2018. *Plants (Basel)* (doi: 10.3390/plants7010013)
- T.K. Mao, J. Van de Water and M.E. Gershwin. 2005. *J Med Food*, **8**, 27-30. (doi: 10.1089/jmf.2005.8.27)
- L.K. Oetjen, M.R. Mack, J. Feng, T.M. Whelan, H. Niu, C.J. Guo, S.V. Chen, A.M. Trier, A.Z. Xu, S.V. Tripathi, J. Luo, X. Gao, L. Yang, S.L. Hamilton, P.L. Wang, J.R. Brestoff, M.L. Council, R. Brasington, A. Schaffer, F. Brombacher, C.S. Hsieh, R.W. Gereau, M.J. Miller, Z.F. Chen, H. Hu, S. Davidson, Q. Liu, B.S. Kim. 2017. *Cell*, **171**, 217-228. (doi: 10.1016/j.cell.2017.08.006)

Tables

No.	Ion mode	r.t. (min)	m/z	VIP	Elemental composition	Metabolites	Control group	Model group ^a	Baicalin group ^b
1	[M-H] ⁻	11.404	303.2386	12.75	C ₂₀ H ₃₂ O ₂	arachidonic acid ^c	747.39±41.24	385.69±21.83**↓	661.68±30.81##↑
2	[M-H] ⁻	11.337	255.2337	10.15	C ₁₆ H ₃₂ O ₂	Palmitic acid ^c	902.98±17.15	656.12±66.22**↓	842.41±44.67##↑
3	[M-H] ⁻	11.202	279.2338	7.39	C ₁₈ H ₃₂ O ₂	Linoleic acid	985.79±48.63	1110.77±77.21**↑	1009.47±62.61##↓
4	[M-H] ⁻	11.045	277.2166	5.73	C ₁₈ H ₃₀ O ₂	γ-Linolenic acid	213.02±29.11	131.22±16.67**↓	193.95±22.25##↑
5	[M-H] ⁻	5.211	178.0521	4.03	C ₉ H ₉ NO ₃	hippuric acid ^c	15.58±2.72	52.38±8.72**↑	25.11±3.01##↓
6	[M-H] ⁻	1.128	89.0264	3.48	C ₃ H ₆ O ₃	lactic acid ^c	3.33±0.23	31.5±6.02**↑	10.13±1.84##↓
7	[M-H] ⁻	10.225	335.2249	2.48	C ₂₀ H ₃₂ O ₄	LTB ₄ ^c	2.64±0.3	16.61±2.33**↑	6.03±0.74##↓
8	[M-H] ⁻	1.121	145.0139	2.39	C ₅ H ₆ O ₅	α- ketoglutaric acid ^c	14.95±2.93	1.74±0.31**↓	11.79±2.16##↑
9	[M-H] ⁻	1.124	191.0121	1.54	C ₆ H ₈ O ₇	citric acid ^c	5.24±0.38	10.57±1.59**↑	6.54±0.55##↓
10	[M-H] ⁻	4.711	181.0522	1.53	C ₉ H ₁₀ O ₄	Hydroxyphenyllactic acid	1.77±0.27	7.09±0.89**↑	3.15±0.21##↓
11	[M-H] ⁻	9.874	257.1752	1.19	C ₁₄ H ₂₆ O ₄	Tetradecanedioic acid	7.58±1.17	4.09±0.65**↓	6.82±0.79##↑
12	[M+H] ⁺	1.116	115.0099	9.34	C ₄ H ₄ O ₄	Fumarate ^c	26.15±3.16	68.32±12.58**↑	37.15±3.8##↓
13	[M+H] ⁺	6.601	147.9719	6.51	C ₄ H ₆ NO	glutamine ^c	96.71±8.88	73.69±3.51**↓	91.79±6.43##↑
14	[M+H] ⁺	1.106	133.0196	6.03	C ₄ H ₆ O ₅	Malate ^c	9.38±1.7	26.95±3.03**↑	13.95±1.01##↓
15	[M+H] ⁺	10.717	544.3403	5.98	C ₂₈ H ₅₀ NO ₇ P	PC(20:4/0:0)	117.21±3.86	97.25±6.23**↓	113.26±3.84##↑
16	[M+H] ⁺	7.601	170.0935	5.84	C ₇ H ₁₁ N ₃ O ₂	3-Methylhistidine	39.66±4.71	58.02±4.77**↑	44.71±3.43##↓
17	[M+H] ⁺	7.632	175.1203	5.03	C ₆ H ₁₄ N ₄ O ₂	arginine ^c	115.13±10.11	133.07±10.41**↑	120.81±8.97#↓
18	[M+H] ⁺	4.581	188.0719	4.25	C ₁₁ H ₉ NO ₂	Indoleacrylic acid	82.11±7.27	69.8±5.43**↓	79.77±5.82##↑
19	[M+H] ⁺	5.610	182.0825	3.46	C ₉ H ₁₁ NO ₃	tyrosine ^c	18.12±1.6	25.16±3.34**↑	20.09±1.34##↓
20	[M+H] ⁺	10.590	317.2118	3.07	C ₂₀ H ₃₀ O ₃	LTA ₄ ^c	0.28±0.05	4.77±0.89**↑	1.43±0.21##↓
21	[M+H] ⁺	0.912	114.0655	2.58	C ₄ H ₇ N ₃ O	creatinine ^c	11.67±1.21	15.19±0.97**↑	12.68±0.92##↓
22	[M+H] ⁺	4.990	132.1023	2.16	C ₆ H ₁₃ NO ₂	leucine ^c	5.44±0.96	2.79±0.46**↓	4.81±0.68##↑
23	[M+H] ⁺	0.831	118.0859	1.95	C ₅ H ₁₁ NO ₂	betaine ^c	24.6±2.98	21.13±2.65**↓	23.95±1.71#↑
24	[M+H] ⁺	7.402	112.0888	1.78	C ₅ H ₉ N ₃	histamine ^c	0.17±0.03	1.7±0.29**↑	0.56±0.08##↓
25	[M+H] ⁺	4.202	154.0979	1.61	C ₇ H ₁₁ N ₃ O	N-Acetylhistamine	0.43±0.05	1.71±0.3**↑	0.76±0.08##↓
26	[M+H] ⁺	6.102	90.0560	1.58	C ₃ H ₇ NO ₂	alanine ^c	5.79±0.66	4.27±0.46**↓	5.46±0.51##↑
27	[M+H] ⁺	6.851	133.0610	1.58	C ₄ H ₈ N ₂ O ₃	asparagine ^c	13.67±0.88	15.48±1.44**↑	14.26±0.83#↓
28	[M+H] ⁺	6.801	106.0521	1.42	C ₃ H ₇ NO ₃	serine ^c	10.1±0.79	8.65±0.79**↓	9.83±0.62##↑
29	[M+H] ⁺	5.411	118.0871	1.01	C ₅ H ₁₁ NO ₂	valine ^c	1.36±0.13	1.92±0.26**↑	1.52±0.09##↓
30	[M+H] ⁺	10.882	496.3403	3.71	C ₂₄ H ₅₀ NO ₇ P	PC(16:0/0:0)	166±4.3	156.03±6.9**↓	165.07±2.75##↑

Compared to control group, * p <0.05, ** p <0.01; Compared to model group, #p<0.05, ## p <0.01; “ ↑ ” means a higher level of metabolites, whereas “ ↓ ” represents a lower level of metabolites. cConfirmed with authentic standards

Figure and table captions

1 Table 1. The information for 30 significantly differential metabolites.

2 Figure 1. The nasal sneezes and scratching numbers of rats in each group. Compared to control group, * p <0.05, ** p
3 <0.01; Compared to model group, #p<0.05, ## p <0.01.

4 Figure 2. The pathological changes of nasal mucosa of rats in each group by histological analysis. A: Control group; B:
5 Model group; C: Baicalin group.

6 Figure 3. Effects of baicalin on IgE, histamine, IL-1 β , IL-4, IL-6 and TNF- α level in serum. A: IgE; B: histamine; C: IL-
7 1 β ; D: IL-4; E: IL-6; F: TNF- α . The data were presented as mean \pm SD, n=10 per group. Compared to control group, * p
8 <0.05, ** p <0.01; Compared to model group, #p<0.05, ## p <0.01.

9 Figure 4. The S-plot analysis and OPLS-DA analysis. A and B: The S-plot analysis between control and model groups for
10 the metabolomics data of different ion mode. C and D: The OPLS-DA analysis among control, model and baicalin groups
11 for the metabolomics data of different ion mode.

12 Figure 5. The metabolic pathway of the altered metabolites analyzed by MetaboAnalyst 4.0.

13
14
15
16
17
18
19
20
21
22
23
24
25
26
27
28
29
30
31
32
33
34
35
36
37
38
39
40
41
42
43
44
45
46
47
48
49
50
51
52
53
54
55
56
57
58
59
60

1
2
3
4
5
6
7
8
9
10
11
12
13
14
15
16
17
18
19
20
21
22
23
24
25
26
27
28
29
30
31
32
33
34
35
36
37
38
39
40
41
42
43
44
45
46
47
48
49
50
51
52
53
54
55
56
57
58
59
60

The nasal sneezes and scratching numbers of rats in each group. Compared to control group, * $p < 0.05$, ** $p < 0.01$; Compared to model group, # $p < 0.05$, ## $p < 0.01$.

196x67mm (300 x 300 DPI)

The pathological changes of nasal mucosa of rats in each group by histological analysis. A: Control group; B: Model group; C: Baicalin group.

196x199mm (300 x 300 DPI)

Effects of baicalin on IgE, histamine, IL-1 β , IL-4, IL-6 and TNF- α level in serum. A: IgE; B: histamine; C: IL-1 β ; D: IL-4; E: IL-6; F: TNF- α . The data were presented as mean \pm SD, n=10 per group. Compared to control group, * p <0.05, ** p <0.01; Compared to model group, #p<0.05, ## p <0.01.

2960x743mm (35 x 35 DPI)

The S-plot analysis and OPLS-DA analysis. A and B: The S-plot analysis between control and model groups for the metabolomics data of different ion mode. C and D: The OPLS-DA analysis among control, model and baicalin groups for the metabolomics data of different ion mode.

329x212mm (300 x 300 DPI)

The metabolic pathway of the altered metabolites analyzed by MetaboAnalyst 4.0.

177x177mm (72 x 72 DPI)

Appendix B

Dear Editors and Reviewers:

Thank you for your letter and the reviewers' comments concerning our manuscript entitled "Metabolomics analysis of baicalin on ovalbumin sensitized allergic rhinitis rats" (ID: RSOS-181081). The comments are all valuable and very helpful for revising and improving our paper, as well as the important guiding significance to our researches. We have studied comments carefully and have made correction which we hope meet with approval. Revised portion are marked in red in the paper. The main corrections in the paper and the responds to the comments are as following:

Responds to the comments:

Associate Editor's comments:

Please ensure that you fully respond to the referee commentary, in particular their observation that the quality of the writing needs to be improved. While we sympathise with the complexities and idiosyncrasy of the English language, you may benefit from the advice of services such as these: <https://royalsociety.org/journals/authors/language-polishing/>.

Response: Now the manuscript is fully edited by American Journal Experts (AJE) and the certify had been upload.

Editor comments:

Thanks for your submission. Please make sure that this manuscript is fully edited by an expert who is a native speaker of English (a very difficult language! sorry). We can only publish the paper if the language is up to the standards of the journal. Best of luck.

Response: Now the manuscript is fully edited by American Journal Experts (AJE) and the certify had been upload.

Reviewer #1:

1. Acroynms should be better defined in the abstract and introduction. E.g. IL-4, SD

Response: Sorry, now corrected.

2. S-plot and OPLS-DA analysis were used in this study, the software package Simca version 14.1 (Umetrics, Sweden) was used to process OPLS-DA analysis, author should defined which software was used to process S-plot analysis, Masslynx or Simca version 14.1.

Response: Sorry, Simca version 14.1 was used to process S-plot

analysis, now added.

3. Author had demonstrated “Reproducibility of the metabolomics was determined from six replicated analyses of the same quality control sample (QC) interspersed throughout the analysis. And the RSD of the peak area of all metabolites are below 20%, which demonstrated good stability and reproducibility” at the part of 4.4 Metabolomics study, the figure of PCA analysis(including QC data) should be provided in the supplemental materials.

Response: Now added.

Special thanks to you for your good comments.

Reviewer #2:

First of all, thank you for your affirmation and suggestions.

1. The research part is well-organized, with clear and detailed step-by-step methods. However, the language quality is not satisfactory and required to be improved. My remarks/queries to the authors are as the attached file.

Response: Considering the Reviewer’s suggestion, the manuscript is fully edited by American Journal Experts (AJE) and the certify had been upload.

Once again, thank you very much for your comments and suggestions.

Thank you and best regards.

Yours sincerely,

Yubin Xu

E-mail: xuyubin1988@126.com